# Antibiotic Use for Common Infections in Pediatric Emergency Departments: A Narrative Review

**DOI:** 10.3390/antibiotics12071092

**Published:** 2023-06-22

**Authors:** Spyridon Karageorgos, Owen Hibberd, Patrick Joseph William Mullally, Roberto Segura-Retana, Shenelle Soyer, Dani Hall

**Affiliations:** 1Faculty of Medicine and Dentistry, Blizard Institute, Queen Mary University of London, London E1 2AT, UK; spiroskarageorgo@gmail.com (S.K.);; 2First Department of Pediatrics, Aghia Sophia Children’s Hospital, National and Kapodistrian University of Athens, 11527 Athens, Greece; 3Emergency Department, Cambridge University Hospitals NHS Foundation Trust, Cambridge CB2 0QQ, UK; 4Department of Medicine, Cardiff University, Cardiff CF10 3AT, UK; 5Pediatric Emergency Department, Hospital Nacional de Niños, San José 0221, Costa Rica; 6Department of Emergency Medicine, Children’s Health Ireland at Crumlin, D12 N512 Dublin, Ireland

**Keywords:** antibiotics, pediatric emergency department, antimicrobial stewardship

## Abstract

Antibiotics are one of the most prescribed medications in pediatric emergency departments. Antimicrobial stewardship programs assist in the reduction of antibiotic use in pediatric patients. However, the establishment of antimicrobial stewardship programs in pediatric EDs remains challenging. Recent studies provide evidence that common infectious diseases treated in the pediatric ED, including acute otitis media, tonsillitis, community-acquired pneumonia, preseptal cellulitis, and urinary-tract infections, can be treated with shorter antibiotic courses. Moreover, there is still controversy regarding the actual need for antibiotic treatment and the optimal dosing scheme for each infection.

## 1. Introduction

Antibiotics represent the most common class of prescribed medication for pediatric patients [1]. When used appropriately, antibiotics can save lives. However, antibiotic overuse has led to an increase in antimicrobial resistance (AMR) that now represents one of the biggest threats to global health [2]. Antibiotics are commonly used in both inpatient and outpatient pediatric settings for various infections. However, a significant proportion of antibiotic use is considered unnecessary [3,4]. Antimicrobial stewardship programs have been developed to promote the judicious use of antibiotics in pediatric patients. A United States study showed that approximately one half of antibiotics prescribed in the emergency department (ED) may be either unnecessary or inappropriate [5]; the ED is an important target for antimicrobial stewardship programs. The challenging ED environment, however, has multiple logistical and provider-level hurdles that antimicrobial stewardship programs need to tackle [6,7].

For this article, antimicrobial use in five of the most common infections seen in the pediatric ED (acute otitis media [8,9], tonsillitis [10], community-acquired pneumonia [11,12], preseptal cellulitis [13,14] and urinary-tract infections [15]) are reviewed. All authors, members of the Don’t Forget the Bubbles (DFTB) team and the DFTB Queen Mary University London Pediatric Emergency MSc program, performed a detailed review of blogs published on the DFTB site (https://dontforgetthebubbles.com/ (accessed between 1 September 2022 to 31 December 2022), guidelines, randomized controlled trials, and cohort studies published in the literature. Areas of controversy (e.g., intravenous versus oral antibiotics, appropriate dose, duration of treatment) are discussed and an evidence-based knowledge synthesis focusing on recent advances regarding the use of antibiotics for each infection in the pediatric emergency department is presented.

## 2. Acute Otitis Media (AOM)

Otitis media (OM) is an infection of the middle ear with approximately 60% of children having at least one episode by the age of 4 years. Most paediatricians, emergency medicine clinicians, and primary-care doctors will see scores of children with otitis media [16,17]. However, despite its frequent presentation, the role of antibiotics is controversial.

Literally meaning ‘inflammation of the middle ear’, otitis media is a general term describing multiple disorders that include acute otitis media (AOM), chronic suppurative otitis media (CSOM), and otitis media with effusion (OME) [9,16]. It occurs as part of the inflammatory process following an upper respiratory-tract infection which, due to the small anatomical space of the middle ear, leads to a cascade of events characterised by negative pressure, exudate, and secretions [16,17]. These make the middle ear an ideal environment for colonization by viral and bacterial organisms. Common viral pathogens include respiratory syncytial virus (RSV), coronaviruses, influenza viruses, adenoviruses, human metapneumovirus, and picornaviruses, whilst common bacterial organisms include *Streptococcus Pneumoniae, Haemophilus influenzae*, and *Moraxella catarrhalis* [16,17].

Otitis media often presents with earache and a low-grade fever [9,16,17]. The diagnosis may be more challenging in younger children when symptoms and signs can be nonspecific: irritability, headache, poor feeding, pulling at the ears, vomiting, or diarrhoea [16,17]. The diagnosis can be made clinically with typical appearances of erythema or bulging of the tympanic membrane on otoscopy. Further investigations are usually not indicated unless an alternative diagnosis, or complication, is suspected [8,16,17].

The treatment of otitis media is mainly symptomatic and aimed at the relief of pain [16,17]. Serious complications are uncommon but can be difficult to treat. Intratemporal and intracranial complications, while exceedingly rare, have significant mortality rates and patients should be admitted for aggressive inpatient management. These serious complications remain rare with or without antibiotic treatment of the primary OM; the number needed to treat to prevent one child from developing mastoiditis is approximately 5000 [16,17,18].

The role of antibiotics is controversial with prescribing practices varying amongst specialities and guidelines varying by country [18,19,20,21,22]. Many antibiotic stewardship programs aim to decrease unnecessary antibiotic use [19,23].

A Cochrane review of 13 randomised control trials (RCTs) of 3401 children compared antibiotics to a placebo [17]. Antibiotics did not reduce the number of children with pain at 24 h, with 60% of children being better regardless of treatment. There was only a slight reduction in the number of children with pain in the subsequent days (the number needed to treat for an additional beneficial outcome of 20) [17]. Although antibiotics were shown to reduce the number of children with tympanic membrane perforations and bilateral AOM, they did not reduce the recurrence rate or hearing loss [17]. The Cochrane review concluded that antibiotics are most beneficial in children younger than two with infection in both ears (likely due to their immature immune systems and shorter, wider, more horizontal, and floppy Eustachian tubes making them more at risk of infection) and in children with suppurative AOM (who are at higher risk of more severe infection and complications) [17,24].

Compared to receiving no antibiotics, parents and caregivers have been shown to have better satisfaction with expectant observation and a delayed course of antibiotics [17,25]. The Cochrane review describes five RCTs of 1149 children comparing outcomes from a delayed course of antibiotics versus immediate antibiotics. There was no observable difference in the number of children with pain at 3 to 14 days, and no difference in hearing loss at 4 weeks, perforations of the eardrum, or late AOM recurrences [17,25]. In the two trials where the use of a delayed prescription was reported, 24% (36/150) and 38% (50/132) of parents and caregivers reported using the delayed prescription at some point during the illness [17,25]. Consequently, many guidelines, including the National Institute of Healthcare Excellence (NICE) and the American Academy of Pediatrics (AAP), recommend expectant observation and a delayed course of antibiotics in most children with AOM, with consideration of immediate antibiotics only if there is suppurative otitis media (otorrhoea, with visible pus in the canal), or if the child is less than two and has bilateral AOM [18,21]. If the child or young person is systemically unwell, has signs of a more serious condition such as mastoiditis or meningitis, or is at high risk of complications (e.g., younger than six months, craniofacial malformations, Trisomy 21,immunodeficiency, cochlear implants, incomplete vaccination status, cancer, or transplant recipient), they should receive immediate antibiotics and be admitted [18,21,26,27,28,29,30].

When antibiotics are given, amoxicillin is the antibiotic of choice due to its high concentration in the middle ear [18,21,24,31]. In cases of penicillin allergy, azithromycin, clarithromycin, or cefuroxime are alternatives [18,21,24]. If there is perforation of the tympanic membrane, ototopical antibiotics such as ofloxacin provide high concentrations of antibiotics without associated side effects [18,21,24]. When antibiotics are indicated, they should be continued for five to seven days [18,21,32]. Improvement should be evident in two to three days [18,21].

A failure to respond to antibiotics may suggest an incorrect diagnosis, the presence of a resistant organism, a viral pathogen, or the development of a complication [16,24]. However, a large trial of 520 children aged six to 23 months with AOM showed a higher rate of treatment failure (defined as worsening of symptoms, incomplete resolution of symptoms, or otoscopic signs of infection at the end of the treatment course) with a five-day course than a longer course [8,33]. Rates of adverse effects and antimicrobial resistance were similar in both short and long courses [33]. This has been incorporated into the AAP guideline, which advises a longer duration of antibiotics in children under two [21,33].

Clinical bottom line (Appendix A).

The above evidence and guidelines demonstrate that the treatment of AOM is primarily symptomatic with analgesia [14,15]. There are high-risk groups in whom antibiotics are most beneficial, however, serious complications remain rare with or without antibiotic treatment [15,16,19,22]. There is also a role for expectant observation and delayed prescription of antibiotics [15,23]. When antibiotics are indicated, amoxicillin is the antibiotic of choice and should be continued for five to seven days with a longer duration being more beneficial in children under two [16,19,26].

## 3. Tonsillitis

Tonsillitis is an inflammation of the tonsils; most cases of tonsillitis are usually viral, but bacterial causes also occur [34]. Throat pain in tonsillitis often occurs suddenly [34,35]. Fever over 38 °C, painful swallowing, tonsillar swelling with exudates, and tender anterior cervical nodes may suggest a bacterial rather than viral cause [34]. Group A β-hemolytic streptococcus (GAβHS) is the most common organism [34,35,36] but a positive swab does not guarantee infection as 8% of children are colonized by GAβHS [37], complicating the decision to treat. Individual signs and symptoms cannot clearly indicate a bacterial cause [34,36,38]. In the postpandemic context, a change in the incidence of invasive disease and antimicrobial sensitivity in general has been observed [39,40,41].

### 3.1. Can Scoring Systems Help to Determine Treatment of Tonsillitis?

Clinical prediction rules (CPR) such as FeverPAIN, Centor, and its modified version McIsaac (Table 1) help to identify patients more likely to have GAβHS infection [10,42] and limit testing for viral causes [38]. Centor can be used from 15 years of age, while FeverPAIN [43] and McIsaac can be used in children over age three years [44]. Each feature is assigned one point; the likelihood of GaβHS increases as the score increases [36,43,45]. However, maximum scores yield only 56% [35,44], 65% [35], and 68% probabilities with Centor, FeverPAIN, and McIsaac respectively [46], while a score under two gives a 15% probability of GaβHS infection [43].

Most guidelines acknowledge using one [36,47,48,49] or more CPRs (Table 2), but there are caveats. Though some studies have vetted the Centor and McIsaac prediction rules [43], they have not all been validated in some populations [35,49] and their use in children [50] and low prevalence areas are limited [10,47]. If these scores are used alone, ruling out, rather than ruling in, GAβHS infection appears more feasible [44,51]. When combined with other measures such as rapid antigen detection tests (RADT), though not all research agree [35,45], the detection of GAβHS improves [47,48,49].

A throat culture is the reference standard for confirming GAβHS throat infection [10]. Even small bacterial colonies can be detected [10,50] but waiting for the culture result delays management decisions [10,50,52].

### 3.2. What Is the Role of Rapid Antigen Detection Tests?

Point-of-care RADTs can rapidly confirm GAβHS with 95% specificity [47] and decrease unnecessary antibiotic prescribing by as much as 25% [10,50,53]. However, if a negative RADT is obtained from school-aged children with tonsillitis symptoms, certain guidelines require confirmation via throat culture, since GAβHS prevalence in this group can reach 37% [10,37,47,48,49,50]. The main disadvantage of RADTs is that they only detect GAβHS and no other pathogens and are dependent on good sampling technique [10,47,48].

### 3.3. How Are Antibiotics Best Used?

Antibiotic rationale against GAβHS infection depends on the prevalence of complications in different regions. Industrialized areas typically have low rates of suppurative and nonsuppurative complications in children [10,35]. Here, the goal is the judicious use of antibiotics to minimize resistance [10,50]. However, in countries such as Australia and India, where the prevalence of acute rheumatic fever is higher, antibiotics are advocated for those with a greater risk of its development (Table 3) [48,49].

Guidelines generally agree that antibiotics should not be offered under two conditions: where CPRs are low indicating a likely viral cause [35,47] or when asymptomatic children are incidentally found to carry the bacteria [35,37]. Patients with maximum scores, most likely to have GAβHS infection, may benefit most from antibiotics [35]. In such patients, UK and Australian guidelines advocate immediate antibiotics [35,49] or suggest confirmation of GAβHS infection via culture [49]. By contrast, US guidelines require laboratory confirmation before prescribing antibiotics in all cases [10,48,55]. Delayed prescriptions can be useful in GaβHS-positive patients and in acute phases where children with mild symptoms have high CPRs [35,47].

Penicillins, macrolides, and cephalosporins are all effective against GAβHS [35,56]. However, macrolide-resistance in GaβHS is increasing following the use of azithromycin in COVID-19 cases for its potential immunomodulatory mechanisms [57,58,59]. GAβHS antibiotic resistance is also expanding to second-line antibiotics [60]. Phenoxymethylpenicillin’s narrow activity against GAβHS [61,62] and lack of resistance [56,62] continues to make it the best-choice first-line antibiotic. Although often seen, the practice of prescribing first-line co-amoxiclav is not recommended [63]. Short courses between five to seven days will treat symptoms [35] and can prevent acute rheumatic fever [64], but some guidelines advocate treating for 10 days to ensure that GAβHS is eradicated to minimize the risk of complication [35,48].

Clinical bottom line (Appendix A).

The evidence shows that the first step of tonsillitis management rests on establishing a bacterial cause, using CPRs and RADTS to increase the detection of GAβHS infection [10,42,47]. Though symptomatic treatment may be considered in many cases [35], the decision to prescribe antibiotics should be weighed against the regional prevalence of complications and individual risk factors [10,35,48,49]. Antibiotic initiation and treatment length are guideline-specific [10,35,48,49,55]; however, phenoxymethylpenicillin remains the first choice globally [35,36,47,48,49] with symptomatic treatment being achieved from five to seven days [35] and GAβHS eradication with 10 days of treatment [35,48].

## 4. Community-Acquired Pneumonia

Community-acquired pneumonia (CAP) is an acute infection of the lung parenchyma acquired outside of the hospital [12,65]. The common etiological agents are predominantly viral, with around one in three cases due to a bacterial pathogen, commonly *Streptococcus pneumoniae* and *Staphylococcus aureus,* whilst atypical infections with *Mycoplasma Pneumoniae* are also seen [12,65,66]. Despite predominantly viral etiologies, traditional management in children has been with extended courses of oral antibiotics. Dogma has been challenged in recent years and the liberal usage of antibiotics in pediatric CAP has been under robust analysis. Four big trials have been published in the last three years [67,68,69,70].

The four-armed double-blinded placebo-controlled CAP-IT study compared three-day courses versus seven-day courses and a low dose of 35–50 mg/kg/day versus 70–90 mg/kg/day of oral amoxicillin [67]. A twice-daily dosing regimen was chosen to improve compliance, different from the standard three-times daily regimens.

Twenty-nine centres within the United Kingdom and Ireland participated, recruiting a total of 824 children between six months and six years of age. Exclusions included underlying respiratory illness, prior β-lactam treatment, contraindications to penicillin including allergy, complicated pneumonia (signs of sepsis, or pleural or parenchymal complications), and bilateral wheeze [67,71].

The study found that three days of treatment was noninferior in treating pediatric CAP compared to longer courses. Lower doses of amoxicillin were also noninferior to higher doses. The time to return to normal activities, development of antimicrobial resistance, compliance, side effects, and additional service use was equivalent between arms. The only benefit of a longer course was a slightly reduced length of cough (12 days versus 10 days) [67,71,72].

Similar findings were replicated in the SAFER noninferiority trial; 281 children between six months and 10 years were block randomized to receive five versus 10 days of high-dose amoxicillin across two centres in Canada in a double-blinded RCT [69].

The primary outcome was a clinical cure for the chest infection requiring no further intervention. Clinical cure was defined as initial improvement in the first four days, improvement in work of breathing, no more than one fever after day four of illness, and a lack of requirement for further treatment or hospital admission up to 21 days. A five-day course was found to be noninferior to a 10-day course. Although per participant analysis at 14 and 21 days (based on the trial arm followed by the participant) did not demonstrate noninferiority, intention to treat analysis (based on the intended randomization arm) demonstrated the noninferiority of shorter courses [69].

In the SCOUT-CAP study, 380 children between six months and six years of age were randomized to either five or 10 days of a penicillin-based antibiotic for treating CAP [68]. This was assessed using a composite outcome called a RADAR score (response adjusted for duration of antibiotic risk). The short-course group had both a 69% higher chance of a more desirable outcome (desired improvement with shorter duration of treatment), and fewer adverse events [68,73].

The presence of genetic mutations responsible for antimicrobial resistance was also assessed between groups. This was done by using bacterial throat swabs to represent lung flora. Shorter courses demonstrated reduced presence of antimicrobial resistance genes on day 19–25. It is unclear whether throat flora is a true representation of lung flora, so these findings may not necessarily mean that shorter courses of antibiotics lead to reduced antimicrobial resistance in the lungs, but this does add to the argument that less is more in CAP prescribing [68,73].

### 4.1. Are Antibiotics Always Required?

The authors of the fourth study, ARCTIC PC, suggested that there may be a limited role for any course of antibiotics in mild pediatric CAP [70].

Four-hundred thirty-two children, between six months and twelve years of age, across fifty-six primary care practices in England were randomized to receive seven days of amoxicillin (50 mg/kg) or seven days of placebo when diagnosed with the uncomplicated lower respiratory-tract infection. This was defined as lasting less than 21 days, being judged by the clinician to have an infective aetiology, and having clinical features such as shortness of breath and exudative cough localizing it to the chest. If there was no clinical certainty, or the patient was deemed to be unwell, then participants were offered enrolment in a parallel observation study [70].

Parents and physicians were asked to rate symptoms on a scale from 1 to 10. The median duration of “bad or worse” symptoms were five days for amoxicillin versus six days for placebo. There was also no significant difference in duration or severity of symptoms in a subgroup analysis of children with chest signs (increased work of breathing, crepitations, or wheeze), fever, a physician rating of being unwell, sputum, or chest rattle or shortness of breath.

The authors argue, therefore, no clear benefit to treating mild lower respiratory-tract infections with amoxicillin [70].

There is also an emerging role of point-of-care ultrasound (POCUS) to detect CAP and guide decision-making. POCUS can potentially add value in detecting radiological changes of pneumoniae and pleural effusions with greater sensitivity than a plain film chest x-ray, which may identify children more likely to benefit from antibiotic treatment [74,75,76].

### 4.2. Are Intravenous Antibiotics Necessary in Children Admitted to Hospital with CAP?

A common assumption among healthcare providers is that intravenous antibiotics are superior to oral antibiotics [77]. Even though this holds true in life-threatening conditions such as sepsis, there is a growing body of evidence that oral administration of antibiotics with high bioavailability, such as amoxicillin and clindamycin, is equally effective for most infections [77,78,79]. In pediatric CAP, there are multiple studies, including two RCTs, comparing oral versus intravenous antibiotic use [80,81,82,83,84,85].

The APPIS trial included 1702 pediatric patients in developing countries, aged three to 59 months old, with severe pneumonia, as per the WHO definition. Children were randomized to receive either oral amoxicillin or intravenous penicillin G. No difference was found in treatment failure between groups (19% in each group, risk difference −0.4% (95%CI −4.2 to 3.3) [81]. The PIVOT multicenter randomized controlled equivalence trial included 246 pediatric patients with WHO-defined pneumonia admitted to eight hospitals in the UK who received oral amoxicillin or intravenous benzylpenicillin. Oral amoxicillin was found to be equally effective in most children [80].

Clinical bottom line (Appendix A).

Historic statements by the Infectious Disease Society of America and the Canadian Pediatric Society imply that much of the guidance for antimicrobial resistance is based on thin evidence [65,86]. When indicated, amoxicillin is the preferred choice of antibiotics; when there is suspicion of an atypical infection (such as Mycoplasma pneumoniae), macrolides can be used in combination with a beta-lactam agent [87,88]. The above studies demonstrate that guidelines recommending 7 to 10-day courses of amoxicillin represent an unnecessarily prolonged course. The argument may also be made that shorter courses may also have a significant effect on antimicrobial stewardship. In children with uncomplicated CAP, who are fully immunized, courses should be short (three to five days) and lower dose if they are to be used at all. Decision-making should also take account of the robustness of the social and cultural safety net given the context of the changing landscape of parental perception on antibiotic treatment, fueled by social media, which may be a driver of hesitancy [89,90,91]. This may improve the rate of adverse effects and aid in antimicrobial stewardship in the management of CAP in children.

## 5. Preseptal Cellulitis

Preseptal cellulitis, also called periorbital cellulitis, is a skin and soft-tissue infection around the eye and anterior to the orbital septum [13,14,92]. The eye itself is not affected. It is often caused by the local spread of infection from minor trauma such as a scratch or insect bite but can also arise from a pre-existing sinusitis or dacryocystitis (inflammation of the lacrimal sac). Causative organisms are *Staphylococcus aureus, Streptococcus pneumoniae, Streptococcus pyogenes,* or, occasionally, *Haemophilus influenza* in the unimmunized. There are now also increasing cases of methicillin-resistant *Staphylococcus aureus* (MRSA) [13,14,92].

Preseptal cellulitis often presents with gradual onset unilateral eyelid oedema, erythema, and a low-grade fever [13,14,92]. Serious complications are rare [93,94,95]. The challenge for clinicians is that the early stages of postseptal cellulitis (orbital cellulitis, involving the eye) can present very similarly [13,96]. Missing a postseptal cellulitis can have devastating consequences such as blindness due to optic-nerve compression, cavernous sinus thrombosis, osteomyelitis, meningitis, or a cerebral abscess [13,14,92]. Clinical features such as high-grade fever, toxic appearance, proptosis, and painful eye movements can help to identify orbital cellulitis, but differentiating between pre- and postseptal cellulitis remains challenging [13,94,96]. The concern for clinicians is that without proper treatment, preseptal cellulitis can progress to postseptal cellulitis [13,14]. For mild or moderate preseptal cellulitis, the diagnosis can be made clinically without investigations [13,97,98]. For more severe infections, blood tests, cultures, and imaging are recommended [13,97,98].

All children with suspected preseptal cellulitis should be treated with antibiotics [13,98]. The antibiotic of choice, however, varies based on local protocols with a lack of consensus agreement on management [98,99]. Due to the rise of MRSA, treatment options include trimethoprim-sulfamethoxazole (TMP-SMX) and clindamycin [13,14]. It should be noted that TMP-SMX does not cover Streptococcus pyogenes; thus, a combination with a beta-lactam may be necessary. The choice of antibiotics is also influenced by immunization status. *Haemophilus influenza* was historically one of the most associated pathogens and causes a more severe invasive infection and so it should be considered in unimmunized or partially immunized children [13,14].

To add to the challenge, there is limited evidence to support whether intravenous antibiotics are better than oral antibiotics, with available evidence being low-to-moderate quality [99,100]. Patients who are over one year of age with mild symptoms can be treated as an outpatient with oral antibiotics. Those with more severe disease or who are less than one year of age should be admitted to the hospital and receive intravenous antibiotics with consideration of switching to oral within 24 to 48 h if improving [13,94,99,100,101]. The location of where children receive intravenous antibiotics is also debatable [99]. The “Intravenous ceftriaxone at home versus intravenous flucloxacillin in hospital for children with cellulitis” (CHOICE) randomised control trial included a subgroup of children with preseptal cellulitis [102]. It found that home treatment with intravenous ceftriaxone was not inferior to hospital treatment with intravenous flucloxacillin, indicating an emerging role for outpatient parenteral antibiotic therapy [102].

A pragmatic view is that all children who are discharged should have a clinical review within 24 to 48 h, either in primary or secondary care [13,14,98]. Some argue that if there are concerns about the reliability or ability to access follow-up, children should be admitted [13,14,97]. For children managed on oral antibiotics, if treatment fails to show improvement after 24 to 48 h, the concern is that the infection is due to resistant organisms or that a complication has developed. These children should be admitted for intravenous antibiotics, imaging, and surgical review for consideration of incision and drainage [13,14,98].

Most cases of preseptal cellulitis will resolve after five to seven days of antibiotics. A longer duration of treatment may be needed if the cellulitis persists or for more severe infections [13,14,98].

Clinical bottom line (Appendix A).

Managing preseptal cellulitis remains challenging for clinicians due to the difficulties in differentiating this from postseptal cellulitis, with limited evidence from studies of low to moderate quality exploring intravenous antibiotics versus oral antibiotics [11,68,70,73,74]. All children with suspected preseptal cellulitis should be treated with antibiotics; options include trimethoprim-sulfamethoxazole (TMP-SMX) and clindamycin to cover for MRSA [11,12]. Pragmatically, children over one year of age who are systemically well can be treated with oral antibiotics for five to seven days, provided there is a clinical review within 24 to 48 h [11,12,72].

## 6. Urinary-Tract Infections

Approximately 5% of children have at least one urinary-tract infection (UTI) before the age of five [103], with the highest incidence in the under ones in both sexes [104]. The most common classification divides UTIs into infection of the upper tract, pyelonephritis, or lower tract, cystitis. However, differentiating between complicated versus uncomplicated infection, rather than infection location, can help decision-making around intravenous versus oral antibiotics and their duration [105]. Complicated UTIs include:Neonates;abdominal and/or bladder mass;kidney and urinary-tract anomalies;urosepsis;organisms other than *Escherichia coli*;atypical clinical course, including the absence of clinical response to an antibiotic within 72 h;renal abscess.

The most common pathogen is *E. coli* in up to 80% of UTI cases [104,106]. Antimicrobial resistance is rising as the prevalence of extended-spectrum beta-lactamase (ESBL) producing bacteria increases, with reports ranging from 0.97–13%, and the prevalence of extensive drug resistance of 0.27–0.9% [107]. Other UTI pathogens include *Klebsiella*, *Proteus,* and *Enterobacter* [104].

### 6.1. Diagnosis

The gold-standard investigation for diagnosis is a positive urine culture, but this is not available for at least 24 h. Consequently, an indirect diagnosis is made with microscopic urine analysis and/or dipstick urinalysis. Due to a lack of standardized values, misdiagnosis of UTI has been reported as high as 50% leading to unnecessary antibiotics in children with subsequent negative cultures [108,109,110]. These are mostly related to a high incidence of unnecessary urine testing in children with no or low clinical suspicion of UTI and the method of urine collection [105]. However, the concern that renal scarring may develop by delaying antibiotics while awaiting culture means most clinicians will start antibiotics based on urinalysis and clinical suspicion of UTI, despite the high rates of false positives [111].

#### 6.1.1. Which Children Need A Urine Sample?

NICE and the European Association of Urology/European Society for Pediatric Urology (EAU/ESPU) guidelines recommend urinalysis and/or culture in any child presenting with fever without obvious cause, or with symptoms suggestive of UTI [112,113]. The AAP guidelines take a similar approach, but with the caveat that these guidelines only apply to children under two years old [114]. Symptoms include dysuria, urinary retention, increased void frequency, or suprapubic pain, as well as nonspecific symptoms such as irritability, poor feeding, lethargy, or abdominal pain.

Different scoring systems, such as UTIcalc, Gorelick, and Duty score, have been developed to help clinicians evaluate the risk of UTI in children and decide whether a urine sample should be collected [115,116,117]. A cross-sectional study evaluated the validity of these scores; both UTIcalc and Gorelick had high sensitivity at 75% and 98% respectively, but low specificity, 16% and 8% respectively, meaning these scores result in a low threshold to collect a urine sample, useful in children under two, where symptoms are usually nonspecific, but urine cultures are often negative. In contrast, the Duty score has a low sensitivity of 8% but high specificity of 99%, meaning it misses high numbers of children with UTIs because it uses variables more specific to symptomatic UTIs, which could be more useful in children older than two years old [118].

#### 6.1.2. How Should Urine Be Collected?

There are different methods to collect a urine sample (Table 4). Suprapubic aspiration and urine catheterization are considered gold standards because of a low risk of contamination, but due to their invasive nature have the potential risk of complication [119]. On the other hand, urine bag collection has a high risk of contamination and is consequently not recommended [114]. Clean catch or midstream samples can be used in children with void control, but still have a moderate risk of contamination. Noninvasive methods are time consuming with a median time to sample of 31 min (IQR 11–66 min) [120]. Recently, the Quick-wee method has been proposed as an alternative, in which stimulation of suprapubic skin for five minutes showed a higher success of a urine sample collection in less time [121]. The risk of invasive methods must be balanced against the risk of contamination and the time to obtain the sample.

Some departments use a two-step approach—collecting urine in a bag and if the sample is positive collecting a second urine via a more invasive approach. It is associated with a reduction in invasive procedures, especially in patients with a low risk of UTI [113].

#### 6.1.3. Interpretation of the Results

Urine dipstick analysis is a simple and low-cost method. The two most useful markers of UTI are leucocyte esterase and nitrites [122,123]. Leucocyte esterase is an indirect marker of pyuria (white cells in urine), which could be due to UTI but could also be due to other conditions such as acute febrile illness, urinary calculi, or Kawasaki disease. Nitrites are breakdown products produced by gram-negative organisms, such as *E. coli*. Both are indirect markers that could suggest a UTI. The sensitivity of leucocytes is between 73–84%, with a specificity of 80–92%, while the sensitivity of nitrites is between 41–57%, with a specificity of 96–99% [122,123]—although leucocytes in the urine may not mean there is a UTI, a lack of leucocytes means a UTI is very unlikely. The opposite holds true for nitrites; a lack of nitrites does not exclude a UTI, but their presence is very suggestive of urinary infection. Importantly, in a recent UK and Ireland multicenter retrospective cohort study, dipstick urinalysis in febrile infants (<90 days) presenting in the ED, showed a moderate sensitivity with a high specificity in diagnosing UTI [124]. The microscopic evaluation may add little as the leukocyte threshold is still controversial and depends on the collection method, with a variation between 5 and 2500 cell/microliter [122]. Likewise, the presence of bacteria on microscopy is controversial, as bacterial presence could be caused by contamination or asymptomatic bacteriuria, rather than infection.

### 6.2. Treatment

Due to the risk of renal scarring, prescribing an empiric antibiotic is recommended in children in whom a UTI is suspected while waiting for culture results. When available, cultures can guide whether antibiotics should be continued or adjusted according to bacterial sensitivity.

Globally, *E. coli* has a low resistance to third-generation cephalosporins and this is why it is the recommended first-line antibiotic in complicated UTIs. Resistance to amoxicillin is reported up to 60%, and so, if used, should be combined with a beta-lactamase inhibitor in the form of amoxicillin-clavulanate [106,112,114]. Other antibiotics, such as trimethoprim/sulfamethoxazole or first-generation cephalosporins, are still recommended by some guidelines, especially for uncomplicated UTIs, although some reports suggest resistance up to 40–50% [125]. Resistance to nitrofurantoin is as low as 0%, and recommended by most guidelines; however, it is not common practice to prescribe it initially because it requires six hourly dosings and only reaches low tissue concentrations. It is therefore only recommended in lower tract infections without fever [125].

#### Intravenous vs. Oral Antibiotics

Most children can be managed with oral antibiotics. Parental antibiotics should be reserved for children with high-risk factors:younger than two to three months;urogenital anatomical alteration (e.g., high-grade RVU, severe bladder dysfunction);complicated infections;unable to tolerate oral therapy;ill-appearing.

Utilization of complementary laboratory studies, especially inflammatory markers, is not useful for making decisions regarding choice of therapy [125].

Recent trials suggest that short antibiotic courses of three to five days can be used in children with cystitis, mainly in older children without fever [126] although not all guidelines support this practice yet. In younger children, particularly those who are febrile, antibiotic duration is recommended to be 7 to 10 days. In children who require parenteral therapy, most can switch to oral therapy after two days if clinically improving [127,128]. In complicated UTIs, treatment up to 14 days should be considered [113,114]. International guidelines vary and are illustrated in Table 5.

Clinical bottom line (Appendix A).

The concern about long-term complications of untreated UTI in children, means many guidelines recommend early treatment [112,113,114,129] before the infection is confirmed by culture, using dipstick analysis to guide diagnosis [119,123,124]. In younger children, a UTI presents with nonspecific symptoms and so clinicians should have a low threshold for obtaining a urine sample [112,113,114,129]. As *E. coli* remains the most common pathogen globally, the antibiotic choice should be guided by local resistance trends [103]. Oral antibiotics are effective in most children, reserving the parenteral route for patients with risk factors [89,90,91,104,105,106]. Short antibiotic courses of between two and four days for lower UTI are advocated by most guidelines, with longer courses for upper UTI [112,113,126,128,129].

## 7. Conclusions

Judicious use of antibiotics in the pediatric ED is of the utmost importance in reducing antimicrobial resistance. As evident from this review, recent studies show that uncomplicated common pediatric infectious diseases, in immunocompetent and vaccinated children, can be treated with shorter antibiotic courses without compromising clinical outcomes. However, vaccine hesitancy and the perpetuation of misinformation through social media reinforce the need for a balanced and individualized approach which considers the wider sociodemographic context. Future studies should focus on identifying potential biomarkers and diagnostic tests differentiating viral versus bacterial infectious diseases in the pediatric ED while optimizing the dose and duration of antibiotic courses. Until more evidence is available, pediatric ED clinicians should focus on reducing antibiotic use by answering three basic questions for every patient in whom an infection is suspected: Do we really need antibiotics? Which dose should we use? How long should we treat for?, as presented in Figure 1.

## Figures and Tables

**Figure 1 antibiotics-12-01092-f001:**
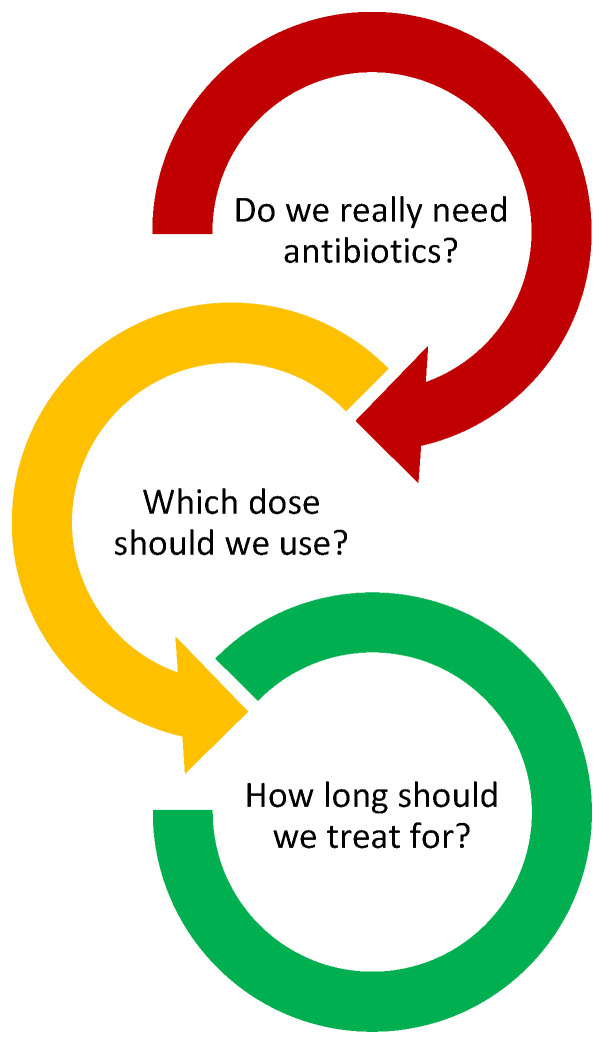
Questions for Pediatric ED physicians.

**Table 1 antibiotics-12-01092-t001:** Modified Centor Prediction Rules (McIsaac) [47].

Tonsillitis Symptom/Parameter	Score
Temperature > 38 °C	1
Absence of cough	1
Anterior cervical adenopathy (tender)	1
Tonsillar swelling or exudates	1
Age 3–14 years	1

**Table 2 antibiotics-12-01092-t002:** A comparison of tonsillitis management by guidelines.

Management Points	IDSA (US) [48]		NICE and SIGN (UK) [35,36]	Rheumatic Heart Disease (RHD) (Australia) [49]	ESCMID (Europe) [47]
Use of clinical scoring systems		Use a clinical score (although no score is specified) to identify patients at low risk of infection, making throat culture and RADT unnecessary	FeverPAIN and Centor to identify children who will benefit more from antibiotics	Centor or Modified Centor to predict children with GAβHS infection	Centor and Modified Centor to predict children with GAβHS infection
	Clinical score validity/utility		Acknowledges no scoring system is validated in the UK population	No clinical scoring system is validated in the Australian population	Advises Centor score has limited use in children
Diagnosis and investigation		Diagnosis on clinical history and exam or clinical score plus RADT and/or throat culture Diagnostic tests not indicated in likely viral aetiology and children < 3 years	Combination of clinical diagnosis, point-of-care testing, and laboratory culture		
	Blood tests	Antistreptococcal (ASO) antibody titres not recommended for acute diagnosis		ASO and anti-DNase B titres can be used to determine recent infection	CRP and procalcitonin not essential for assessment
	RADTs	Positive RADTs do not need confirmation by culture. Negative RADTs in children and adolescents should be confirmed by throat culture	RADTs in children with high FeverPAIN scores confers no benefit to using clinical score alone	RADTs not commonly used as they are not as accurate as culture	RADTs are 95% specific for GAβHS when compared to throat culture (90% sensitivity). Should be considered in children with high Centor scores to increase RADT accuracy
	Throat culture	Selective throat culture testing to avoid identifying carriers rather than infection	Throat swabs should not be used routinely	Throat swabs in those with high Centor or McIsaac scores	Throat culture unnecessary, especially if RADT is negative
Antibiotic treatment		Antibiotics should only be prescribed for proven GAβHS infection	Antibiotics are more effective in children with a positive throat swab	Give antibiotic treatment for GAβHS positive swabs	Balance benefits of antibiotics in high Centor scores and positive GAβHS against effects on microbiome, resistance, side effects, and cost
	Empirical treatment		Antibiotic prophylaxis is not endorsed for recurrent sore throat	Empirical antibiotics only in those at high risk for acute rheumatic failure, regardless of symptoms or tests	
	Immediate antibiotic prescription		Immediate antibiotics should be used in systemic infection, serious illness, or high risk of complications		
	Delayed prescription		Consider delayed prescription with FeverPAIN of 4 or Centor of 3		Delayed prescriptions can be used
Antibiotic rationale		To prevent suppurative and nonsuppurative complications (acknowledges that development of poststreptococcal glomerulonephritis is unaffected by antibiotic treatment)	Antibiotics should not be used to reduce complications, or routinely decrease community cross infection	To prevent the occurrence of acute rheumatic fever	Antibiotics are not indicated to prevent complications in low-risk patients
First-line antibiotic		Phenoxymethylpenicillin (oral), Or amoxicillin (oral), Or benzathine benzylpenicillin G (intramuscular)	Phenoxymethylpenicillin (oral)	Benzathine benzylpenicillin G (intramuscular), Or phenoxymethylpenicillin (oral)	Phenoxymethylpenicillin (oral) Amoxicillin can be used in younger children, but not recommended in older
Allergy to first-line antibiotics		Cephalexin (oral), cefadroxil (oral), clindamycin (oral), azithromycin (oral), clarithromycin (oral)	Clarithromycin (oral)	Hypersensitivity: cephalexin (oral) Anaphylaxis: azithromycin (oral)	No recommended alternative
Duration of antibiotic	Short course	Shorter antibiotic courses of oral cephalosporins are not endorsed	Use the shortest effective course Short antibiotic courses achieve symptomatic cure	Neither long nor short antibiotic durations are mentioned	Insufficient evidence to endorse antibiotic courses shorter than 10 days
	Long course (10 days)	Long courses are needed for maximum GAβHS eradication	Long course achieves microbiological cure		

IDSA: Infectious Diseases Society of America; NICE: National Institute of Healthcare Excellence; SIGN: Scottish Intercollegiate Guidelines Network ESCMID: European Society of Clinical Microbiology and Infectious Diseases; RADT: Rapid Antigen Detection Test, GAβHS: Group A beta-hemolytic streptococcus.

**Table 3 antibiotics-12-01092-t003:** Risk factors for developing acute rheumatic fever [49,54].

Living in an ARF endemic setting
Previous/Recent Family history of ARF or RHD
Prior GABHS infection (throat or skin)
Regular travel to ARF endemic setting
Peak age for ARF development (5–20 years)
Limited household resources Overcrowding Cold, damp environment Bathing, laundry facilities
Poor access to medical facilities
Refugee or Migrant status from low- or middle-income country (RHD)

ARF: Acute Rheumatic Fever; RHD: Rheumatic Heart Disease; GAβHS: Group A beta-haemolytic streptococcus.

**Table 4 antibiotics-12-01092-t004:** Urine collection methods, contamination rate and complications [119].

Method	Contamination Rate	Complications
Suprapubic aspiration	0–7%	Hematuria (3.6%) Risk of aspiration of gut lumen (1/140 procedures) Failure at first attempt 10–54%
Urethral catheterisation	14.3%	Microscopic hematuria (17%) Risk of septicemia in neonates (not defined) Failure at first attempt < 10%
Clean catch	16–38%	Time consuming Moderately high contamination rate
Bag collected	43.9–88%	Time consuming High contamination rate

**Table 5 antibiotics-12-01092-t005:** International pediatric UTI guidelines.

Guidelines	NICE (2007) [112]	AAP (2011) [114]	EUA (2021) [113]	KHA-CARI (2015) [129]
Population	0–16 years old	3 months–2 years old	0–18 years old	0–18 years old
Diagnosis				
Urine sample	CCU MSU Alternative: Collection bag SPA or BC only when other methods are not possible	BC SPA	CCU MSU BC SPA	SPA BC CCU MSU
Two-step approach	Not mentioned	Recommended	Recommended	Recommended
Culture confirmation	Only if dipstick is not conclusive	Always	Always	Always
Bacterial confirmation	Not described	Positive urine culture BC: 5 × 10^4^ CFU/mL SPA: any growth	Positive urine culture BC: >10^3–5^ CFU/mL CCU: >10^4^ CFU/mL + symptoms SPA: any growth	Positive urine culture BC or CCU: > 10^8^ CFU/L SPA: any growth
Treatment				
Route	IV: <3 months, NPO or unwell Oral: all other children	IV: NPO or unwell Oral: all other children	IV: <3 months, NPO or unwell Oral: all other children	IV: <3 months, NPO or unwell Oral: all other children
Duration	Lower UTI: 3 days Upper UTI: 7–10 days	7–14 days	Lower UTI: at least 3–5 days Upper UTI: 7–14 days	Lower UTI: 2–4 days Upper UTI: 7–10 days

BC: bladder catheterization; CCU: Clean catch urine; MSU: Midstream urine; SPA: Suprapubic aspiration; CFU: colonies formation units; NPO: nil per os; IV: intravenous. NICE: National Institute of Healthcare Excellence; AAP: American Association of Pediatrics; EUA: European Urology Association: KHA-CARI: Kidney Health Australia.

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
