# Peer review of "Antibiotic Use for Common Infections in Pediatric Emergency Departments: A Narrative Review"

_antibiotics, 2023, doi:10.3390/antibiotics12071092_

Round 1

Reviewer 1 Report

Overall, this manuscript was very well-written and has good scientific merit.

Names of viral pathogens (e.g. respiratory syncytial virus, coronavirus) need not be italicized.

A common practice (particularly in developing nations) is the prescription on co-amoxyclav instead of just amoxycillin for tonsillitis even though there has been no convincing evidence of beta-lactamase production by GABHS to date. This malpractice should also be addressed if it is rampant in the authors’ country.

For CAP, the role of macrolides (to cover for atypical pneumonia pathogens such as Mycoplasma pneumoniae) in combination with a beta-lactam agent should be addressed as well.

Species names (i.e. aureus, pneumoniae and pyogenes) mentioned in the ‘Pre-septal cellulitis’ section do not need to be capitalized.

If TMP-SMX is used for empirical treatment of pre-septal cellulitis, it should be stated that this antibiotic does not cover for Strep. pyogenes. A combination with a beta-lactam agent may be needed.

The authors state that “E. coli has a low resistance to third generation cephalosporins and this is why it is the recommended first-line antibiotic in uncomplicated UTIs”. Is there a mistake in the phrasing of this statement? If this was true, there will be a limited role for oral antibiotics in such UTIs since 3rd gen cephalosporins are not administered orally.

Author Response

Overall, this manuscript was very well-written and has good scientific merit.

We thank the reviewer for their comments and suggestions.

Names of viral pathogens (e.g. respiratory syncytial virus, coronavirus) need not be italicized.

We have revised the manuscript accordingly.

A common practice (particularly in developing nations) is the prescription on co-amoxyclav instead of just amoxycillin for tonsillitis even though there has been no convincing evidence of beta-lactamase production by GABHS to date. This malpractice should also be addressed if it is rampant in the authors’ country.

For CAP, the role of macrolides (to cover for atypical pneumonia pathogens such as Mycoplasma pneumoniae) in combination with a beta-lactam agent should be addressed as well.

Species names (i.e. aureus, pneumoniae and pyogenes) mentioned in the ‘Pre-septal cellulitis’ section do not need to be capitalized.

Thank you for detecting these. Updated.

If TMP-SMX is used for empirical treatment of pre-septal cellulitis, it should be stated that this antibiotic does not cover for Strep. pyogenes. A combination with a beta-lactam agent may be needed.

Revised and now reads as “It should be noted that TMP-SMX does not cover Streptococcus pyogenes, thus a combination with a beta-lactam may be necessary.”

The authors state that “E. coli has a low resistance to third generation cephalosporins and this is why it is the recommended first-line antibiotic in uncomplicated UTIs”. Is there a mistake in the phrasing of this statement? If this was true, there will be a limited role for oral antibiotics in such UTIs since 3rd gen cephalosporins are not administered orally.

Thank you for the comment. Indeed, there was a typo. It has been updated and now reads as follows:

“E. coli has a low resistance to third generation cephalosporins and this is why it is the recommended first-line antibiotic in complicated UTIs”.

Other antibiotics as trimethoprim/sulfamethoxazole or first-generation cephalosporins are still recommended by some guidelines, especially for uncomplicated UTIs,”

Reviewer 2 Report

Well done. Thank you for your submission

There are a few capitalization errors throughout the paper, mainly with bacterial names. 

Author Response

Well done. Thank you for your submission.

There are a few capitalization errors throughout the paper, mainly with bacterial names.

We thank reviewer 2 for the positive comments.

We have updated the errors accordingly.

Reviewer 3 Report

Dear colleagues, thank you so much for this excellent paper!

Here are some minor areas that could be addressed in order to improve current review

In page 3 you are stating " If the child or young person is systemically unwell, has signs of a more serious condition such as mastoiditis or meningitis, or is at high risk of complications (younger than six months, cranio-facial malformations, Trisomy 21, and immunodeficiency), they should receive immediate antibiotics and be admitted [18,21]."

You cad add to present list of risk factors some others :

1. children with cochlear implants 

(Davidson L, Foley DA, Clifford P, Blyth CC, Bowen AC, Hazelton B, Kuthubutheen J, McLeod C, Rodrigues S, Tay SM, Campbell AJ. Infectious complications and optimising infection prevention for children with cochlear implants. J Paediatr Child Health. 2022 Jun;58(6):1007-1012. doi: 10.1111/jpc.15889. Epub 2022 Feb 9. PMID: 35138003. 

and 

Vila PM, Ghogomu NT, Odom-John AR, Hullar TE, Hirose K. Infectious complications of pediatric cochlear implants are highly influenced by otitis media. Int J Pediatr Otorhinolaryngol. 2017 Jun;97:76-82. doi: 10.1016/j.ijporl.2017.02.026. Epub 2017 Mar 12. PMID: 28483256; PMCID: PMC6198317.]

a. especially those with incomplete vaccination status 

(Rose O, Moriarty R, Brown C, Teagle H, Neeff M. Vaccination rates in cochlear implant patients: a review of paediatric recipients. J Laryngol Otol. 2022 Jul;136(7):628-631. doi: 10.1017/S0022215121003133. Epub 2021 Oct 26. PMID: 34698004.)

b. or those with AOM in a child that had a removal procedure of ventilation tubes, before cochlear implantation

(Raveh E, Ulanovski D, Attias J, Shkedy Y, Sokolov M. Acute mastoiditis in children with a cochlear implant. Int J Pediatr Otorhinolaryngol. 2016 Feb;81:80-3. doi: 10.1016/j.ijporl.2015.12.016. Epub 2015 Dec 31. PMID: 26810295.

2. children with cancer or transplant [they can be included in the broad group of "immunodeficiency" but I think it is worthy to point-up to these increasing groups of children, even in relative low-resource settings] (Restuti RD, Sriyana AA, Priyono H, Saleh-Saleh RR, Airlangga TJ, Zizlavsky S, Suwento R, Yasin FH. Chronic suppurative otitis media and immunocompromised status in paediatric patients. Med J Malaysia. 2022 Sep;77(5):619-621. PMID: 36169076.

Table 1 is not user-friendly and duplicates information presented above. Mainly because of potential confusion between "no antibiotic" and "consider delayed prescription" sections of table.

Page 7 Treatment section for GAS tonsillitis

Some aspects need to be emphasized in current post-pandemic epidemiological situation :

1. Macrolide-resistance in GAS is increasing and potentially will further increase because of the massive over-prescription of azithromycin in COVID-19 cases, for immunomodulatory mechanisms    

Powell LM, Choi SJ, Chipman CE, Grund ME, LaSala PR, Lukomski S. Emergence of Erythromycin-Resistant Invasive Group A Streptococcus, West Virginia, USA, 2020-2021. Emerg Infect Dis. 2023 May;29(5):898–908. doi: 10.3201/eid2905.221421. PMID: 37080963; PMCID: PMC10124663.

Brouwer S, Rivera-Hernandez T, Curren BF, Harbison-Price N, De Oliveira DMP, Jespersen MG, Davies MR, Walker MJ. Pathogenesis, epidemiology and control of Group A Streptococcus infection. Nat Rev Microbiol. 2023 Mar 9:1–17. doi: 10.1038/s41579-023-00865-7. Epub ahead of print. PMID: 36894668; PMCID: PMC9998027.

Echeverría-Esnal D, Martin-Ontiyuelo C, Navarrete-Rouco ME, De-Antonio Cuscó M, Ferrández O, Horcajada JP, Grau S. Azithromycin in the treatment of COVID-19: a review. Expert Rev Anti Infect Ther. 2021 Feb;19(2):147-163. doi: 10.1080/14787210.2020.1813024. Epub 2020 Oct 6. PMID: 32853038.

2. GAS antibiotic-resistance is expanding to second-line antibiotics or to antibiotics that are not indicated for GAS tonsillitis (like vancomycin)

Tadesse M, Hailu Y, Biset S, Ferede G, Gelaw B. Prevalence, Antibiotic Susceptibility Profile and Associated Factors of Group A Streptococcal pharyngitis Among Pediatric Patients with Acute Pharyngitis in Gondar, Northwest Ethiopia. Infect Drug Resist. 2023 Mar 22;16:1637-1648. doi: 10.2147/IDR.S402292. PMID: 36992964; PMCID: PMC10040342.

3. Change of incidence of invasive disease produced by pathogens of upper airway, in post-pandemic context

Brueggemann AB, Jansen van Rensburg MJ, Shaw D, McCarthy ND, Jolley KA, Maiden MCJ, van der Linden MPG, Amin-Chowdhury Z, Bennett DE, Borrow R, Brandileone MC, Broughton K, Campbell R, Cao B, Casanova C, Choi EH, Chu YW, Clark SA, Claus H, Coelho J, Corcoran M, Cottrell S, Cunney RJ, Dalby T, Davies H, de Gouveia L, Deghmane AE, Demczuk W, Desmet S, Drew RJ, du Plessis M, Erlendsdottir H, Fry NK, Fuursted K, Gray SJ, Henriques-Normark B, Hale T, Hilty M, Hoffmann S, Humphreys H, Ip M, Jacobsson S, Johnston J, Kozakova J, Kristinsson KG, Krizova P, Kuch A, Ladhani SN, Lâm TT, Lebedova V, Lindholm L, Litt DJ, Martin I, Martiny D, Mattheus W, McElligott M, Meehan M, Meiring S, Mölling P, Morfeldt E, Morgan J, Mulhall RM, Muñoz-Almagro C, Murdoch DR, Murphy J, Musilek M, Mzabi A, Perez-Argüello A, Perrin M, Perry M, Redin A, Roberts R, Roberts M, Rokney A, Ron M, Scott KJ, Sheppard CL, Siira L, SkoczyÅ„ska A, Sloan M, Slotved HC, Smith AJ, Song JY, Taha MK, Toropainen M, Tsang D, Vainio A, van Sorge NM, Varon E, Vlach J, Vogel U, Vohrnova S, von Gottberg A, Zanella RC, Zhou F. Changes in the incidence of invasive disease due to Streptococcus pneumoniae, Haemophilus influenzae, and Neisseria meningitidis during the COVID-19 pandemic in 26 countries and territories in the Invasive Respiratory Infection Surveillance Initiative: a prospective analysis of surveillance data. Lancet Digit Health. 2021 Jun;3(6):e360-e370. doi: 10.1016/S2589-7500(21)00077-7. Erratum in: Lancet Digit Health. 2021 May 26;: PMID: 34045002; PMCID: PMC8166576.

4. Post-pandemic change of antimicrobial susceptibility in community-acquired infections, in general 

Smith DRM, Shirreff G, Temime L, Opatowski L. Collateral impacts of pandemic COVID-19 drive the nosocomial spread of antibiotic resistance: A modelling study. PLoS Med. 2023 Jun 5;20(6):e1004240. doi: 10.1371/journal.pmed.1004240. PMID: 37276186; PMCID: PMC10241372.

Yousafzai ADK, Bangash AH, Asghar SY, Abbas SMM, Khawaja HF, Zehra S, Khan AU, Kamil M, Ayesha N, Khan AK, Mohsin R, Ahmed O, Fatima A, Ali A, Badar AU, Abbasi MN, Ashraf M, Shah AH, Iqbal T. Clinical efficacy of Azithromycin for COVID-19 management: A systematic meta-analysis of meta-analyses. Heart Lung. 2023 Jul-Aug;60:127-132. doi: 10.1016/j.hrtlng.2023.03.004. Epub 2023 Mar 16. PMID: 36996755; PMCID: PMC10017380.

In page 9, CAP section you state "In children with uncomplicated CAP, courses should be short (three to five days) and lower dose if they are to be used at all. This may improve the rate of adverse effects, and aid in antimicrobial stewardship in the management of CAP in children."

This statement needs a small nuance, because in low-resource settings, in populations with low coverage of childhood immunizations and in a changing landscape of parental perception on antibiotic treatment, fueled by Social Media and other hesitancy drivers, CAP is still "Enemy nr 1" generating excess-mortality, beyond neonatal age.

1. Van Hecke O, Butler CC, Wang K, Tonkin-Crine S. Parents' perceptions of antibiotic use and antibiotic resistance (PAUSE): a qualitative interview study. J Antimicrob Chemother. 2019 Jun 1;74(6):1741-1747. doi: 10.1093/jac/dkz091. PMID: 30879040; PMCID: PMC6524473.

2. Shamim MA, Padhi BK, Satapathy P, Siddiq A, Manna S, Aggarwal AK, Al-Ahdal T, Khubchandani J, Henao-Martinez AF, Sah R. Parents' expectation of antibiotic prescriptions for respiratory infections in children: a systematic review and meta-analysis. Ther Adv Infect Dis. 2023 May 12;10:20499361231169429. doi: 10.1177/20499361231169429. PMID: 37206057; PMCID: PMC10189844.

3. Dantuluri KL, Bonnet KR, Schlundt DG, Schulte RJ, Griffith HG, Luu A, Charnogursky C, Perkins JM, Whitmore CC, Banerjee R, Howard LM, Grijalva CG. Antibiotic perceptions, adherence, and disposal practices among parents of pediatric patients. PLoS One. 2023 Feb 9;18(2):e0281660. doi: 10.1371/journal.pone.0281660. PMID: 36758043; PMCID: PMC9910628.

A brief comment on point-of-care ultrasound imaging for CAP. Could add some value in detecting patients that need antibiotic treatment, with a greater accuracy than radiology in finding small pleural effusions. Pediatric lung-ultrasound scoring systems can increase proper selection of treatment. 

1. Bloise S, Marcellino A, Sanseviero M, Martucci V, Testa A, Leone R, Del Giudice E, Frasacco B, Gizzone P, Proietti Ciolli C, Ventriglia F, Lubrano R. Point-of-Care Thoracic Ultrasound in Children: New Advances in Pediatric Emergency Setting. Diagnostics (Basel). 2023 May 17;13(10):1765. doi: 10.3390/diagnostics13101765. PMID: 37238249; PMCID: PMC10217038.

2. Yan JH, Yu N, Wang YH, Gao YB, Pan L. Lung ultrasound vs chest radiography in the diagnosis of children pneumonia: Systematic evidence. Medicine (Baltimore). 2020 Dec 11;99(50):e23671. doi: 10.1097/MD.0000000000023671. PMID: 33327356; PMCID: PMC7738074.

3. Ciuca IM, Dediu M, Pop LL. Pediatric pneumonia (PedPne) lung ultrasound score and inflammatory markers: A pilot study. Pediatr Pulmonol. 2022 Feb;57(2):576-582. doi: 10.1002/ppul.25760. Epub 2021 Nov 23. PMID: 34786878.

Page 12 Table 5 has a strange aspect of the right section - maybe if you use justify alignment would improve aspect?!

Page 14 Conclusion section

You are stating "As evident from this review, recent studies show that common pediatric infectious diseases can be treated with shorter antibiotic courses without compromising clinical outcome..." needs some nuances.

a. Current status of vaccine hesitancy (Larson, H.J. Defining and measuring vaccine hesitancy. Nat Hum Behav 6, 1609–1610 (2022). https://doi.org/10.1038/s41562-022-01484-7) and parental perception on recommended versus optional vaccines (Miron VD, Toma AR, Filimon C, Bar G, Craiu M. Optional Vaccines in Children-Knowledge, Attitudes, and Practices in Romanian Parents. Vaccines (Basel). 2022 Mar 7;10(3):404. doi: 10.3390/vaccines10030404. PMID: 35335036; PMCID: PMC8955643.)

b. disinformation spread among parents via Social Media (Aïmeur E, Amri S, Brassard G. Fake news, disinformation and misinformation in social media: a review. Soc Netw Anal Min. 2023;13(1):30. doi: 10.1007/s13278-023-01028-5. Epub 2023 Feb 9. PMID: 36789378; PMCID: PMC9910783.

Desai AN, Ruidera D, Steinbrink JM, Granwehr B, Lee DH. Misinformation and Disinformation: The Potential Disadvantages of Social Media in Infectious Disease and How to Combat Them. Clin Infect Dis. 2022 May 15;74(Suppl_3):e34-e39. doi: 10.1093/cid/ciac109. PMID: 35568471; PMCID: PMC9384020.

Pool J, Fatehi F, Akhlaghpour S. Infodemic, Misinformation and Disinformation in Pandemics: Scientific Landscape and the Road Ahead for Public Health Informatics Research. Stud Health Technol Inform. 2021 May 27;281:764-768. doi: 10.3233/SHTI210278. PMID: 34042681.),

c. the unknown role of AI (chat bots like ChatGPT) in an environment without regulatory limits or safety features

(Lee P, Bubeck S, Petro J. Benefits, Limits, and Risks of GPT-4 as an AI Chatbot for Medicine. N Engl J Med. 2023 Mar 30;388(13):1233-1239. doi: 10.1056/NEJMsr2214184. PMID: 36988602.)

d. polarization of society both in socio-economical aspects and medical access will impact this approach on common infection treatment...

Thank you again for a stimulating paper.

Author Response

Dear colleagues, thank you so much for this excellent paper!

Here are some minor areas that could be addressed in order to improve current review

In page 3 you are stating " If the child or young person is systemically unwell, has signs of a more serious condition such as mastoiditis or meningitis, or is at high risk of complications (younger than six months, cranio-facial malformations, Trisomy 21, and immunodeficiency), they should receive immediate antibiotics and be admitted [18,21]."

You cad add to present list of risk factors some others :

  1. children with cochlear implants 

(Davidson L, Foley DA, Clifford P, Blyth CC, Bowen AC, Hazelton B, Kuthubutheen J, McLeod C, Rodrigues S, Tay SM, Campbell AJ. Infectious complications and optimising infection prevention for children with cochlear implants. J Paediatr Child Health. 2022 Jun;58(6):1007-1012. doi: 10.1111/jpc.15889. Epub 2022 Feb 9. PMID: 35138003. 

and 

Vila PM, Ghogomu NT, Odom-John AR, Hullar TE, Hirose K. Infectious complications of pediatric cochlear implants are highly influenced by otitis media. Int J Pediatr Otorhinolaryngol. 2017 Jun;97:76-82. doi: 10.1016/j.ijporl.2017.02.026. Epub 2017 Mar 12. PMID: 28483256; PMCID: PMC6198317.]

  1. especially those with incomplete vaccination status 

(Rose O, Moriarty R, Brown C, Teagle H, Neeff M. Vaccination rates in cochlear implant patients: a review of paediatric recipients. J Laryngol Otol. 2022 Jul;136(7):628-631. doi: 10.1017/S0022215121003133. Epub 2021 Oct 26. PMID: 34698004.)

  1. or those with AOM in a child that had a removal procedure of ventilation tubes, before cochlear implantation

(Raveh E, Ulanovski D, Attias J, Shkedy Y, Sokolov M. Acute mastoiditis in children with a cochlear implant. Int J Pediatr Otorhinolaryngol. 2016 Feb;81:80-3. doi: 10.1016/j.ijporl.2015.12.016. Epub 2015 Dec 31. PMID: 26810295.) 

  1. children with cancer or transplant [they can be included in the broad group of "immunodeficiency" but I think it is worthy to point-up to these increasing groups of children, even in relative low-resource settings] (Restuti RD, Sriyana AA, Priyono H, Saleh-Saleh RR, Airlangga TJ, Zizlavsky S, Suwento R, Yasin FH. Chronic suppurative otitis media and immunocompromised status in paediatric patients. Med J Malaysia. 2022 Sep;77(5):619-621. PMID: 36169076.) 

Thank you for this excellent and well referenced suggestion. The text has been updated in line with this comment and now reads as follows:

“If the child or young person is systemically unwell, has signs of a more serious condition such as mastoiditis or meningitis, or is at high risk of complications (younger than six months, cranio-facial malformations, Trisomy 21, and immunodeficiency, cochlear implants, incomplete vaccination status, cancer, transplant), they should receive immediate antibiotics and be admitted.”

Table 1 is not user-friendly and duplicates information presented above. Mainly because of potential confusion between "no antibiotic" and "consider delayed prescription" sections of table.

Thank you. This has now been removed to avoid repetition and potential for confusion.

Page 7 Treatment section for GAS tonsillitis

Some aspects need to be emphasized in current post-pandemic epidemiological situation:

  1. Macrolide-resistance in GAS is increasing and potentially will further increase because of the massive over-prescription of azithromycin in COVID-19 cases, for immunomodulatory mechanisms    

Powell LM, Choi SJ, Chipman CE, Grund ME, LaSala PR, Lukomski S. Emergence of Erythromycin-Resistant Invasive Group A Streptococcus, West Virginia, USA, 2020-2021. Emerg Infect Dis. 2023 May;29(5):898–908. doi: 10.3201/eid2905.221421. PMID: 37080963; PMCID: PMC10124663.

Brouwer S, Rivera-Hernandez T, Curren BF, Harbison-Price N, De Oliveira DMP, Jespersen MG, Davies MR, Walker MJ. Pathogenesis, epidemiology and control of Group A Streptococcus infection. Nat Rev Microbiol. 2023 Mar 9:1–17. doi: 10.1038/s41579-023-00865-7. Epub ahead of print. PMID: 36894668; PMCID: PMC9998027.

Echeverría-Esnal D, Martin-Ontiyuelo C, Navarrete-Rouco ME, De-Antonio Cuscó M, Ferrández O, Horcajada JP, Grau S. Azithromycin in the treatment of COVID-19: a review. Expert Rev Anti Infect Ther. 2021 Feb;19(2):147-163. doi: 10.1080/14787210.2020.1813024. Epub 2020 Oct 6. PMID: 32853038.

Thank you. This has been added to the text and now reads:

“Penicillins, macrolides and cephalosporins are all effective against GAβHS. However, macrolide-resistance in GaβHS is increasing following the use of azithromycin in Covid-19 cases for its potential immunomodulatory mechanisms”.

  1. GAS antibiotic-resistance is expanding to second-line antibiotics or to antibiotics that are not indicated for GAS tonsillitis (like vancomycin)

Tadesse M, Hailu Y, Biset S, Ferede G, Gelaw B. Prevalence, Antibiotic Susceptibility Profile and Associated Factors of Group A Streptococcal pharyngitis Among Pediatric Patients with Acute Pharyngitis in Gondar, Northwest Ethiopia. Infect Drug Resist. 2023 Mar 22;16:1637-1648. doi: 10.2147/IDR.S402292. PMID: 36992964; PMCID: PMC10040342.

Thank you. This has been added to the text and now reads:

“Penicillins, macrolides and cephalosporins are all effective against GAβHS. However, macrolide-resistance in GaβHS is increasing following the use of azithromycin in Covid-19 cases for its immunomodulatory mechanisms. GaβHS antibiotic resistance is also expanding to second-line antibiotics. Phenoxymethylpenicillin’s narrow activity against GaβHS and lack of resistance continues to make it the best choice first-line antibiotic”.

  1. Change of incidence of invasive disease produced by pathogens of upper airway, in post-pandemic context

Brueggemann AB, Jansen van Rensburg MJ, Shaw D, McCarthy ND, Jolley KA, Maiden MCJ, van der Linden MPG, Amin-Chowdhury Z, Bennett DE, Borrow R, Brandileone MC, Broughton K, Campbell R, Cao B, Casanova C, Choi EH, Chu YW, Clark SA, Claus H, Coelho J, Corcoran M, Cottrell S, Cunney RJ, Dalby T, Davies H, de Gouveia L, Deghmane AE, Demczuk W, Desmet S, Drew RJ, du Plessis M, Erlendsdottir H, Fry NK, Fuursted K, Gray SJ, Henriques-Normark B, Hale T, Hilty M, Hoffmann S, Humphreys H, Ip M, Jacobsson S, Johnston J, Kozakova J, Kristinsson KG, Krizova P, Kuch A, Ladhani SN, Lâm TT, Lebedova V, Lindholm L, Litt DJ, Martin I, Martiny D, Mattheus W, McElligott M, Meehan M, Meiring S, Mölling P, Morfeldt E, Morgan J, Mulhall RM, Muñoz-Almagro C, Murdoch DR, Murphy J, Musilek M, Mzabi A, Perez-Argüello A, Perrin M, Perry M, Redin A, Roberts R, Roberts M, Rokney A, Ron M, Scott KJ, Sheppard CL, Siira L, SkoczyÅ„ska A, Sloan M, Slotved HC, Smith AJ, Song JY, Taha MK, Toropainen M, Tsang D, Vainio A, van Sorge NM, Varon E, Vlach J, Vogel U, Vohrnova S, von Gottberg A, Zanella RC, Zhou F. Changes in the incidence of invasive disease due to Streptococcus pneumoniae, Haemophilus influenzae, and Neisseria meningitidis during the COVID-19 pandemic in 26 countries and territories in the Invasive Respiratory Infection Surveillance Initiative: a prospective analysis of surveillance data. Lancet Digit Health. 2021 Jun;3(6):e360-e370. doi: 10.1016/S2589-7500(21)00077-7. Erratum in: Lancet Digit Health. 2021 May 26;: PMID: 34045002; PMCID: PMC8166576.

Thank you. This has been updated and now reads as follows:

“In the post-pandemic context, a change in the incidence of invasive disease and antimicrobial sensitivity in general has been observed.”

  1. Post-pandemic change of antimicrobial susceptibility in community-acquired infections, in general 

Smith DRM, Shirreff G, Temime L, Opatowski L. Collateral impacts of pandemic COVID-19 drive the nosocomial spread of antibiotic resistance: A modelling study. PLoS Med. 2023 Jun 5;20(6):e1004240. doi: 10.1371/journal.pmed.1004240. PMID: 37276186; PMCID: PMC10241372.

Yousafzai ADK, Bangash AH, Asghar SY, Abbas SMM, Khawaja HF, Zehra S, Khan AU, Kamil M, Ayesha N, Khan AK, Mohsin R, Ahmed O, Fatima A, Ali A, Badar AU, Abbasi MN, Ashraf M, Shah AH, Iqbal T. Clinical efficacy of Azithromycin for COVID-19 management: A systematic meta-analysis of meta-analyses. Heart Lung. 2023 Jul-Aug;60:127-132. doi: 10.1016/j.hrtlng.2023.03.004. Epub 2023 Mar 16. PMID: 36996755; PMCID: PMC10017380.

Thank you. This has been updated and now reads as follows:

“In the post-pandemic context, a change in the incidence of invasive disease and antimicrobial sensitivity in general has been observed.”

In page 9, CAP section you state "In children with uncomplicated CAP, courses should be short (three to five days) and lower dose if they are to be used at all. This may improve the rate of adverse effects, and aid in antimicrobial stewardship in the management of CAP in children."

This statement needs a small nuance, because in low-resource settings, in populations with low coverage of childhood immunizations and in a changing landscape of parental perception on antibiotic treatment, fueled by Social Media and other hesitancy drivers, CAP is still "Enemy nr 1" generating excess-mortality, beyond neonatal age.

Thank you. This has been updated and now reads as follows:

“In children with uncomplicated CAP, who are fully immunized, courses should be short (three to five days) and lower dose if they are to be used at all. Decision making should also take account of the robustness of the social and cultural safety net given the context of the changing landscape of parental perception on antibiotic treatment, fueled by social media, which may be a driver of hesitancy. This may improve the rate of adverse effects, and aid in antimicrobial stewardship in the management of CAP in children.”

  1. Van Hecke O, Butler CC, Wang K, Tonkin-Crine S. Parents' perceptions of antibiotic use and antibiotic resistance (PAUSE): a qualitative interview study. J Antimicrob Chemother. 2019 Jun 1;74(6):1741-1747. doi: 10.1093/jac/dkz091. PMID: 30879040; PMCID: PMC6524473.
  2. Shamim MA, Padhi BK, Satapathy P, Siddiq A, Manna S, Aggarwal AK, Al-Ahdal T, Khubchandani J, Henao-Martinez AF, Sah R. Parents' expectation of antibiotic prescriptions for respiratory infections in children: a systematic review and meta-analysis. Ther Adv Infect Dis. 2023 May 12;10:20499361231169429. doi: 10.1177/20499361231169429. PMID: 37206057; PMCID: PMC10189844.
  3. Dantuluri KL, Bonnet KR, Schlundt DG, Schulte RJ, Griffith HG, Luu A, Charnogursky C, Perkins JM, Whitmore CC, Banerjee R, Howard LM, Grijalva CG. Antibiotic perceptions, adherence, and disposal practices among parents of pediatric patients. PLoS One. 2023 Feb 9;18(2):e0281660. doi: 10.1371/journal.pone.0281660. PMID: 36758043; PMCID: PMC9910628.

A brief comment on point-of-care ultrasound imaging for CAP. Could add some value in detecting patients that need antibiotic treatment, with a greater accuracy than radiology in finding small pleural effusions. Pediatric lung-ultrasound scoring systems can increase proper selection of treatment. 

Thank you. This has been updated and now reads as follows:

“There is also an emerging role of point-of-care ultrasound (POCUS) to detect CAP and guide decision making. POCUS can potentially add value in detecting radiological changes of pneumoniae and pleural effusions, with greater sensitivity than plain film chest x-ray, which may identify children more likely to benefit from antibiotic treatment. Pediatric lung-ultrasound scoring system can further increase appropriate selection of children for antibiotic treatment.”

  1. Bloise S, Marcellino A, Sanseviero M, Martucci V, Testa A, Leone R, Del Giudice E, Frasacco B, Gizzone P, Proietti Ciolli C, Ventriglia F, Lubrano R. Point-of-Care Thoracic Ultrasound in Children: New Advances in Pediatric Emergency Setting. Diagnostics (Basel). 2023 May 17;13(10):1765. doi: 10.3390/diagnostics13101765. PMID: 37238249; PMCID: PMC10217038.
  2. Yan JH, Yu N, Wang YH, Gao YB, Pan L. Lung ultrasound vs chest radiography in the diagnosis of children pneumonia: Systematic evidence. Medicine (Baltimore). 2020 Dec 11;99(50):e23671. doi: 10.1097/MD.0000000000023671. PMID: 33327356; PMCID: PMC7738074.
  3. Ciuca IM, Dediu M, Pop LL. Pediatric pneumonia (PedPne) lung ultrasound score and inflammatory markers: A pilot study. Pediatr Pulmonol. 2022 Feb;57(2):576-582. doi: 10.1002/ppul.25760. Epub 2021 Nov 23. PMID: 34786878.

Page 12 Table 5 has a strange aspect of the right section - maybe if you use justify alignment would improve aspect?!

Thank you. Alignment has been justified; further changes can be made if required.

Page 14 Conclusion section

You are stating "As evident from this review, recent studies show that common pediatric infectious diseases can be treated with shorter antibiotic courses without compromising clinical outcome..." needs some nuances.

Thank you. This has been updated and now reads as follows:

“Judicious use of antibiotics in the pediatric ED is of utmost importance in reducing antimicrobial resistance. As evident from this review, recent studies show that uncomplicated common pediatric infectious diseases, in immunocompetent and vaccinated children, can be treated with shorter antibiotic courses without compromising clinical outcomes. However, vaccine hesitancy and the perpetuation of misinformation through social media reinforce the need for a balanced and individualized approach which considers the wider socio-demographic context.”

  1. Current status of vaccine hesitancy (Larson, H.J. Defining and measuring vaccine hesitancy. Nat Hum Behav6, 1609–1610 (2022). https://doi.org/10.1038/s41562-022-01484-7) and parental perception on recommended versus optional vaccines (Miron VD, Toma AR, Filimon C, Bar G, Craiu M. Optional Vaccines in Children-Knowledge, Attitudes, and Practices in Romanian Parents. Vaccines (Basel). 2022 Mar 7;10(3):404. doi: 10.3390/vaccines10030404. PMID: 35335036; PMCID: PMC8955643.)
  2. disinformation spread among parents via Social Media (Aïmeur E, Amri S, Brassard G. Fake news, disinformation and misinformation in social media: a review. Soc Netw Anal Min. 2023;13(1):30. doi: 10.1007/s13278-023-01028-5. Epub 2023 Feb 9. PMID: 36789378; PMCID: PMC9910783.

Desai AN, Ruidera D, Steinbrink JM, Granwehr B, Lee DH. Misinformation and Disinformation: The Potential Disadvantages of Social Media in Infectious Disease and How to Combat Them. Clin Infect Dis. 2022 May 15;74(Suppl_3):e34-e39. doi: 10.1093/cid/ciac109. PMID: 35568471; PMCID: PMC9384020.

Pool J, Fatehi F, Akhlaghpour S. Infodemic, Misinformation and Disinformation in Pandemics: Scientific Landscape and the Road Ahead for Public Health Informatics Research. Stud Health Technol Inform. 2021 May 27;281:764-768. doi: 10.3233/SHTI210278. PMID: 34042681.),

  1. the unknown role of AI (chat bots like ChatGPT) in an environment without regulatory limits or safety features

(Lee P, Bubeck S, Petro J. Benefits, Limits, and Risks of GPT-4 as an AI Chatbot for Medicine. N Engl J Med. 2023 Mar 30;388(13):1233-1239. doi: 10.1056/NEJMsr2214184. PMID: 36988602.)

  1. polarization of society both in socio-economical aspects and medical access will impact this approach on common infection treatment...

Thank you again for a stimulating paper.

Thank you again for a thorough review and suggestions.